# A computational study of co-inhibitory immune complex assembly at the interface between T cells and antigen presenting cells

**Zhaoqian Su, Kalyani Dhusia, Yinghao Wu** *

Department of Systems and Computational Biology, Albert Einstein College of Medicine, NY, United States of America

* yinghao.wu@einstein.yu.edu

## Abstract

The activation and differentiation of T-cells are mainly directly by their co-regulatory receptors. T lymphocyte-associated protein-4 (CTLA-4) and programed cell death-1 (PD-1) are two of the most important co-regulatory receptors. Binding of PD-1 and CTLA-4 with their corresponding ligands programed cell death-ligand 1 (PD-L1) and B7 on the antigen presenting cells (APC) activates two central co-inhibitory signaling pathways to suppress T cell functions. Interestingly, recent experiments have identified a new *cis*-interaction between PD-L1 and B7, suggesting that a crosstalk exists between two co-inhibitory receptors and the two pairs of ligand-receptor complexes can undergo dynamic oligomerization. Inspired by these experimental evidences, we developed a coarse-grained model to characterize the assembling of an immune complex consisting of CLTA-4, B7, PD-L1 and PD-1. These four proteins and their interactions form a small network motif. The temporal dynamics and spatial pattern formation of this network was simulated by a diffusion-reaction algorithm. Our simulation method incorporates the membrane confinement of cell surface proteins and geometric arrangement of different binding interfaces between these proteins. A wide range of binding constants was tested for the interactions involved in the network. Interestingly, we show that the CTLA-4/B7 ligand-receptor complexes can first form linear oligomers, while these oligomers further align together into two-dimensional clusters. Similar phenomenon has also been observed in other systems of cell surface proteins. Our test results further indicate that both co-inhibitory signaling pathways activated by B7 and PD-L1 can be down-regulated by the new *cis*-interaction between these two ligands, consistent with previous experimental evidences. Finally, the simulations also suggest that the dynamic and the spatial properties of the immune complex assembly are highly determined by the energetics of molecular interactions in the network. Our study, therefore, brings new insights to the co-regulatory mechanisms of T cell activation.

## Author summary

The activation of a T cell can be regulated by the receptors on its surface, such as CTLA-4 and PD-1. People used to think that these two receptors inhibit T cell activation through

**Data Availability Statement:** All relevant data are within the manuscript and its Supporting Information files.

**Funding:** Funding was received for this work by YW from the National Institutes of Health under

Grant Numbers R01GM120238 and
R01GM122804. The work is also partially
supported by a start-up grant from Albert Einstein
College of Medicine. The funders had no role in
study design, data collection and analysis, decision
to publish, or preparation of the manuscript.

**Competing interests:** The authors have declared
that no competing interests exist.

distinct pathways. However, recent experiments discovered that the ligands of these two
receptors, B7 and PD-L1, can interact with each other on the same surface of antigen pre-
senting cells. Here we utilized computational simulations to investigate functional roles of
this newly discovered interaction in T cell coregulation. The specific environment of
interface between T cell and antigen presenting cell has been taken into account of our
model. Ligand and receptors randomly diffuse within this interface area. They further
involve in different types of interactions, with each other from the same side or the oppo-
site side of cell surface. Using this method, we found ligands and receptors can not only
form complexes, but also aggregate into large-scale clusters. We also demonstrated that
the engagement between B7 and PD-L1 can reduce the interactions with their corre-
sponding receptors. This study, therefore, offers new insights to our understanding of sig-
nal regulation in T cells.

## Introduction

The activation of T cells is induced after the engagement of T cell receptors (TCR) with major
histocompatibility complexes (MHC) presented on the surfaces of antigen presenting cells
(APC) [1]. This original activating signal can be modified either positively or negatively by the
co-regulatory receptors, which are also membrane proteins on the surface of T cell [2]. The
negative modification of signals is conducted by co-inhibitory receptors such as cytotoxic T
lymphocyte-associated protein-4 (CTLA-4) and programed cell death-1 (PD-1) [3], while the
positive modification is carried out by co-stimulatory receptors including CD28 and 4-1BB
[4]. Accumulating evidences showed that these receptors can modulate the immune system in
a manner which is more complicated than we used to believe. For instance, the B7-1/B7-2:
CD28/CTLA-4 pathway is the best characterized system for T-cell co-regulation [5]. Binding
of B7-1 or B7-2 ligands to CD28 or CTLA-4 receptors respectively triggers the co-stimulatory
or co-inhibitory signaling pathways [6]. Even within this well-studied system, new interactions
continue to be identified. It was previously thought that B7 ligand does not bind to other B7
homologues such as programed cell death-ligand 1 (PD-L1) [7]. However, recent experiments
demonstrated the existence of this interaction, resulting in a bi-directional inhibitory signal
which is context-dependent [8,9].

   Moreover, it has been discovered that these coregulatory receptors and their ligands often
contain more than one binding interfaces. The binding-interface between receptors and
ligands leads to the formation of so-called "*trans*-interaction" which links T cell to APC [10–
13]. In addition to this traditional binding interface, a secondary binding interface exists
between receptors or ligands on the same cell surfaces. Interactions through this type of inter-
faces are called "*cis*-interactions" [14]. For instance, B7 and PD-L1 can form *trans*-interactions
with their corresponding receptors, CTLA-4 and PD-1. As mentioned above, B7 and PD-L1
was recently found to interact with each other. More recent study further indicated that this
interaction is formed on the same cell surface [15], and is through an interface that is partially
overlapped with their other interfaces involved in dimerization or *trans*-interaction [16].
While the intersecting and competing interactions among co-regulatory receptors and their
ligands result in a complex signaling network, it is likely that the combination of their *trans*
and *cis*-interactions further drive the aggregation of these ligand-receptor complexes into
higher-order oligomers at the interface between T cell and APC. As observed in previous
experiments, co-regulatory receptors such as CTLA-4 and CD28 are spatially co-localized
together with TCR/MHC complexes after the initial encounter of T cells with APC, leading

into the assembly of signaling hubs called micro-clusters [17]. After further spreading of the T cell/APC interface, these micro-clusters merge together into a ring-like structure called central supramolecular activation cluster (cSMAC), as part of the immunological synapse [18]. Like many other systems of ligand-receptor complexes [19], formation of these spatial-temporal patterns at the interface between T cells and APC plays an essential role as the intercellular platform to regulate the dynamics of immune signaling pathways.

Unfortunately, the mechanisms underlying the spatial organization of networks formed by ligand-receptor interactions and its functional implication in T cell co-regulation remain far from being fully understood, partially due to the complexity of the system and the limitations in current experimental approaches. High-throughput screening techniques such as yeast-two-hybrid have been proven less effective to discover new interactions between proteins associated with cell membrane [20]. Moreover, it is difficult to derive structures of membrane proteins and their interactions by experimental methods such as X-ray crystallography [21] and NMR spectroscopy [22]. As a result, some interactions in the network of immune co-regulation and the binding interfaces in their relevant proteins have only been revealed very recently. On the other hand, it is also highly challenging not only to determine the dynamic properties of interactions between membrane proteins, but also to detect their subsequent oligomerization on cell surfaces. *In vitro* methods such as surface plasma resonance (SPR) measure protein binding within an unrealistic environment comparing with the membrane surface of living cells [23], while *in vivo* fluorescence-based microscopy is restricted by the spatial and temporal resolution and requires careful development of the appropriate labels for target proteins [24–28]. Alternatively, numerical simulations allow scientists to test hypothesis that might not be currently approachable in the laboratory. As a result, a large variety of computational models have been developed to study ligand-receptor interactions at cellular interfaces [29–38].

Based on recently observed experimental evidences, here we use a small network motif as a test system to study the complexity in the co-regulation of immune signaling pathways. Four elements of the network and their relationship are illustrated in **Fig 1A**. These elements form two pairs of *trans*-interactions (CTLA-4/B7 and PD-1/PD-L1) at the interface between T cell and APC as classic regulators of T cell co-inhibitory pathways. The newly discovered *cis*-interaction between B7 and PD-L1 was further integrated into the network, leading into additional crosstalk between two original signaling components. The temporal dynamics and spatial pattern formation of this network was studied by coarse-grained simulations. Our simulation method incorporates the membrane confinement of cell surface proteins and geometric arrangement of different binding interfaces between these proteins. A wide range of binding constants was tested for the interactions involved in the network. In summary, the test results indicate that the dynamic balance between the two *trans*-interactions is closely regulated by their *cis*-interaction in the network. We further show that large-scale clusters can be formed by ligand-receptor complexes through the combination of these *trans* and *cis*-interactions. Our study, therefore, sheds light on the co-regulatory mechanisms of T cell activation, while the method can be potentially applied to simulate the spatial organization in other systems at cellular interfaces.

## Model and methods

### Description of the test system

The test system of this study is a simplified network motif which contains two co-inhibitory receptors and two ligands. As shown in **Fig 1A**, there are four different types of interactions in the network. The first interaction is formed between receptor CTLA-4 and its ligand B7, while the second is the interaction between receptor PD-1 and its ligand PD-L1. The next interaction is between two B7 proteins, which leads to their homo-dimerization, as observed in the x-ray

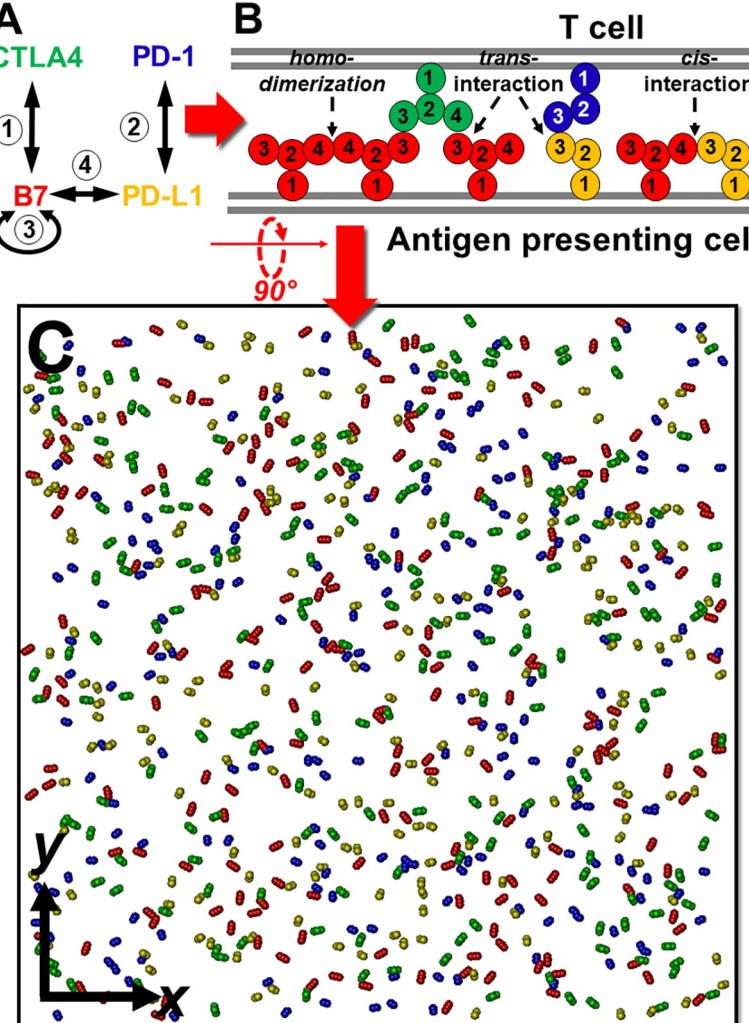

**Fig 1. A simple network motif of T cell coregulation is used as a test system to study immune complex assembly.**
In specific, four proteins are involved in the network. Their interactions are shown in (**A**). These receptors and ligands contain different types of binding interfaces and are located at the interface between T cell and antigen presenting cell (APC), as described by our coarse-grained model (**B**). Based on this model representation, a diffusion-reaction algorithm is applied to simulate the system. The simulation starts from an initial configuration, the top view of which is illustrated in (**C**). CTLA-4 is shown in green, B7 is shown in red, PD-L1 is shown in orange, and PD-1 is shown in blue.

crystallographic structure. The last interaction in the network is recently discovered between B7 and PD-L1. As a result, B7 is directly involved in three different types of interactions, therefore becomes the hub of the network. It is worth mentioning that this four-element network motif is the minimalist system of co-regulatory complex. The B7 family consists of at least seven cell surface ligands [39]. B7-1 and B7-2 are the two major types in the family. These two ligands share their receptor CTLA-4, but can bind to other receptors such as CD28. However, the binding of CD28 to B7-1 and B7-2 is outcompeted by CTLA-4 due to its substantially lower binding affinities to both ligands [40]. The competition between CTLA-4 and CD28 and the differentiation of members in the B7 family are not considered in current study. The incorporation of these factors into our simulations will add another layer of complexity to the spatial organization of coregulatory signaling, which is beyond the scope of this study.

## Representation of the spatial model

As shown in **Fig 1A**, some proteins in the test system have more than one binding partners. Some of these binding partners are mutually exclusive in a protein complex because they have overlapping binding interfaces. On the other hand, some other binding partners can simultaneously exist in a same protein complex. Moreover, some of interactions in the network are formed between proteins on the surfaces of different cells, called *trans*-interactions, while some other interactions are formed between proteins on the same cell surfaces, called *cis*-interactions. The locations of proteins on different cell types and the spatial arrangements of their different binding sites are incorporated into our model as an intercellular region sketched in **Fig 1B**. Receptor CTLA-4 and PD-1 are expressed on T cell as the top side of the region, while ligands B7 and PD-L1 are expressed on APC as the bottom side of the region.

The structure of each protein is further represented by different numbers of inter-connected groups. The distance and geometry among groups in a protein are fixed. Based on previous experimental evidences in the crystal structure of the human CTLA-4/B7-1 co-stimulatory complex (PDB id 1I8L) [41], CTLA-4 exists as homodimers on T cells through an interface of highly conserved residues. Each CTLA-4 in the dimer further binds to a B7 monomer simultaneously, providing the structural basis of a zipper-like oligomerization. Because the homo-dimerization of CTLA-4 does not interfere with any other interactions in this study, we assume that CTLA-4 dimers preform in the system to avoid further complication. As a result, 4 groups are used to describe the structure of CTLA-4. While the first 2 group represent the extracellular region of the dimer, the 3rd and 4th groups indicate the two symmetric binding sites in each monomer. As the ligand of CTLA-4, B7 is presented on APC by 4 groups. The first two groups are used to represent its extracellular region. The 3rd group represents the binding site which can form the *trans*-interaction with either binding site in a CTLA-4 dimer. The 4th group of B7 represents the binding site for its homo-dimerization, given the structural evidence that the *trans*-interaction and the homo-dimerization of B7 are formed through non-overlapping interfaces. In parallel, the structures of both PD-1 and its ligand PD-L1 are represented by 3 groups. Their extracellular regions are represented by the first two groups, while the 3rd groups in both proteins regulate their *trans*-interaction. Finally, the 3rd group of PD-L1 can also form a *cis*-interaction with the 4th group of B7, based on the experimental evidence that this *cis*-interaction partially shares binding interface with B7 homo-dimerization and PD-1/PD-L1 *trans*-interaction.

## Algorithm of the diffusion-reaction simulation

Based on the model representation for each protein and the description of interaction between different proteins, an initial configuration can be constructed, as shown in **Fig 1C**. The interface between T cell and APC is modeled as two layers of flat surfaces overlapping on top of each other. The size of each surface was variated from 300nm×300nm to 700nm×700nm to adjust protein's surface densities, and the distance between two surfaces (20nm) is a typical value of intercellular distance observed experimentally [42]. A large amount of both receptors CTLA-4 and PD-1 are randomly placed on top bound of surface, while their ligands form random distributions on the opposite side of surface layer. The height of each receptor and ligand equals 10nm, corresponding to the typical size of their extracellular domains. Given the experimentally measured surface density of receptors on T cells [43], the numbers of CTLA-4 and PD-1 in the system are variated within the range of $\sim10^2$ molecules. The same range of surface density is applied to the ligands B7 and PD-L1.

Following above initial configuration, the spatial-temporal dynamics of the system is evolved by a diffusion-reaction simulation algorithm [44,45]. Specifically, there are two

separate scenarios within each time step. In the first scenario, all proteins are selected by random order to make stochastic diffusions. Different from the free diffusions of soluble proteins in three-dimensional space and along three rotational degrees of freedom, diffusions of surface-anchored receptors and ligands are confined. As a result, rotations of all proteins are restricted along the membrane normal with only one degree of freedom, while their translational movements are limited within the two-dimensional plane of cell surface. The periodic boundary condition is further applied to both surface layers along their x and y directions. After diffusions are carried out for all proteins, the kinetics of their binding reactions is simulated in the second scenario. Association between two proteins is triggered if the distance between the two corresponding binding sites is below a predetermined distance cutoff. A rate constant $k_{on}$ is further given to determine the probability of association. Association can be classified into following categories: 1) occurred between a receptor and a ligand through their *trans*-binding sites, 2) occurred between two B7 proteins through their homo-dimerization sites, or 3) occurred between B7 and PD-L1 through their *cis*-binding sites. A single binding site cannot interact with more than one binding site simultaneously.

After the formation of a *trans*-interaction, the ligand-receptor complex diffuses together at the cellular interface with a relatively lower diffusion coefficient. In contrast, we assume that if the interaction is formed between two proteins on the same cell surface, they will stop diffusing. This assumption was made to facilitate lateral oligomerization of ligand-receptor complexes. It is based on the following consideration. Previous single molecule imaging experiment showed that diffusions of membrane proteins on the same cell surface can be reduced by at least an order of magnitude upon oligomerization [46]. Moreover, diffusions were even slower as the size of oligomers became larger. It was proposed that these oligomers of ligands or receptors formed through the *cis*-interactions are trapped due to enhanced connections between their intracellular regions and cytoskeleton. Therefore, although the mobility of entire oligomers was still detectable in the experiment, they are set as static in our simulation to avoid computational complexity. We believe this simplification will not significantly affect our simulation outputs.

In addition to association, its reverse process is also modeled as dissociation for any interaction which was formed in the previous simulation steps. The probability of dissociation is determined by the rate constant $k_{off}$. After dissociation of two specific proteins, they can either re-associate as a geminate recombination if their distance is still below the cutoff, or diffuse farther away from each other. The iteration of above diffusion-reaction process will be terminated after the dynamics of the simulated system reaches equilibrium.

### Determination of simulation parameters

The timescale for each simulation step is set as 1ns. The distance cutoff that is used to trigger protein-protein association equals 0.5nm. The two-dimensional diffusion constants of a monomeric receptor or ligand were derived from our previous study [47], in which all-atom molecular dynamic (MD) simulations were carried out for membrane receptors on the lipid bilayer [48]. In specific, the translational diffusion constant equals 10μm$^2$/s and rotational diffusion coefficient equals 1˚ per ns. Moreover, we assume that the diffusions of a ligand-receptor complex at cell interface are even slower, with a translational constant of 5μm$^2$/s and rotational coefficient of 0.28˚ per ns. All association rates in the system are fixed to a value that makes the time-scale of our simulations computationally accessible. This value corresponds to an experimentally measurable $k_{on}$ on the level of $10^9 M^{-1} s^{-1}$. This is at the upper bound of a diffusion-limited rate constant, in which association is accelerated by long-range electrostatic interactions [49]. On the other side, the dissociation rates $k_{off}$ is determined by $k_{off} = k_{on} \times \exp$

$(-\Delta G_0/kT)$ in which $\Delta G_0$ is the binding affinity of a protein-protein interaction. These binding affinities were selected from the typical range of interactions between membrane receptors and ligands on the surfaces of immune cells, corresponding to the dissociation constants from millimolar (mM) to nanomolar (nM) [50].

## Results

### The CTLA-4/B7 complexes form two-dimensional clusters at cell interface

Before the simulation of dynamics in the full network as shown in **Fig 1A**, we first started from a simpler system which contains a pair of interacting proteins: the receptor CTLA-4 and its ligand B7. As we introduced in the **Model and Methods**, CTLA-4 receptors preexist as symmetric homodimers. Each of the two binding sites in a homodimer can form a *trans*-interaction with B7, while B7 themselves can further homo-dimerize together, as shown in **Fig 1B**. As an initial test, 200 CTLA-4 receptor dimers were randomly placed on the surface of T cell, while another 200 B7 were distributed on the opposite side of APC surface. The simulation box has the dimension of 500nm×500nm×20nm. Receptors and ligands then underwent stochastic diffusions and complex formation at the interface between T cell and APC, which algorithm has been described in the **Model and Methods**. The binding affinity of the *trans*-interaction between CTLA-4 and B7 was set to be -7kT in the initial test. The same binding affinity was also used to regulate the homo-dimerization between B7.

Given these parameters, a simulation with the total length of $1\times10^7$ ns was carried out. The kinetics profiles of protein-protein interactions are plotted in **Fig 2A**. The number of *trans*-interactions between CTLA-4 and B7 is shown by the blue curve as a function of time in the figure, while the number of homodimers formed between B7 is shown by the red curve. Some representative snapshots selected from the simulation trajectory were also plotted. Receptors CTLA-4 in these snapshots are shown in green with four representative groups, whereas ligands B7 are shown in red by the same representation. Soon after the initial random distribution (**Fig 2B**), small linear oligomers were observed in the system. These oligomers consist of both CTLA-4 and B7 through their *trans*-interaction and homo-dimerization, as shown in **Fig 2C**. Interestingly, these linear oligomers can grow into two-dimensional clusters when the simulation proceeded (**Fig 2D**). The clustering is a very dynamic process, as small clusters either disappeared or merged into neighboring larger clusters. As a result, when the system reached equilibrium after about $5\times10^6$ ns, a final stable configuration was obtained in which most ligands and receptors aggregated into a single large cluster, as shown in **Fig 2E**. Because each CTLA-4 contains two binding sites, this higher binding avidity cause more *trans*-interactions (>150) than the number of homodimers (~50) at the end of the simulation.

The detailed structure of a linear oligomer in **Fig 2C** has further been enlarged in **Fig 2F**. The figure shows that the oligomer consists of a homodimer of two B7 ligands in the middle and two CTLA-4 receptors at both ends through their *trans*-interactions with B7. Our simulation suggests that this initial oligomerization of ligand-receptor complexes provides a seeding process to extend their length or grow into two-dimensional clusters. Moreover, the detailed structure of a two-dimensional cluster in **Fig 2D** has also been enlarged in **Fig 2G**. The figure indicates that the cluster is composed of multiple linear oligomers which pack against each other. We speculate that the initial formation of a linear oligomer blocks the diffusions of nearby ligands or receptors at cellular interface. These molecules at the proximity of the oligomer therefore are more likely to form complexes and further aggregate into parallel arrays as shown in the figure. Interestingly, the packing of linear assemblies into larger two-dimensional structures at intercellular region has recently also been observed in the system of protocadherin [51–53].

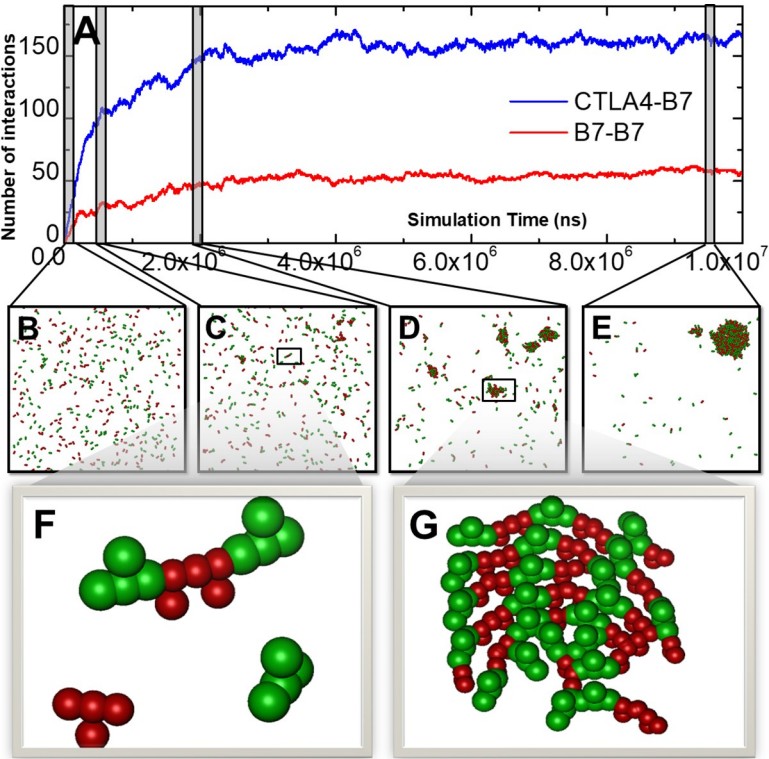

**Fig 2. We started our simulations from a simple system which only contains receptor CTLA-4 and its ligand B7.** The kinetics profiles of protein-protein interactions in an initial test are plotted in (**A**) as a function of simulation time. Some representative snapshots were selected along the simulation trajectory. The same color code and representation are used as before. The initial configuration is shown in (**B**). The simulation shows that small linear oligomers (**C**) can grow into two-dimensional clusters (**D**). These small clusters finally merged into a single large cluster (**E**). The close-up view of a linear oligomer and a two-dimensional cluster are further plotted in (**F**) and (**G**), respectively.

In order to investigate the correlation between CTLA-4/B7 *trans*-interaction and B7 homo-dimerization on a more systematical level, we simultaneously changed the binding affinities in both types of interactions from 0 to -13kT. A total number of 7×7 = 49 combinations were tested. Same as above, the simulation of each combination contains 200 receptors and 200 ligands. The overall results are illustrated as two-dimensional heat maps. Different colors in the maps indicate the number of CTLA-4/B7 *trans*-interactions (**Fig 3A**) and the number of B7-B7 dimers (**Fig 3B**), respectively. The values of two binding affinities are indexed along x axis and y axis. The figures suggest that the *trans*-interaction and homo-dimerization are correlated with each other in the systems. More specifically, instead of the horizontal distribution which indicates no correlation, the color contours in the map of **Fig 3A** shift towards the bottom side in the middle, but then change directions by shifting towards the top side under the strong affinities of B7 dimerization. Similarly, instead of the perpendicular distributions, the color contours in the map of **Fig 3B** shift towards the left-hand side in the middle, but then change directions by shifting towards the right-hand side under the strong affinities of CTLA-4/B7 *trans*-interaction.

These results suggest that the *trans*-interactions and homo-dimerization can mutually strengthen each other when their binding affinities are in the medium range. This effect of positive coupling will be diminished, however, under strong affinities. In order to understand the underlying mechanism of this phenomenon, we plotted the final configurations for some representative areas in the contour maps. **Fig 3C** shows the configuration under strong homo-

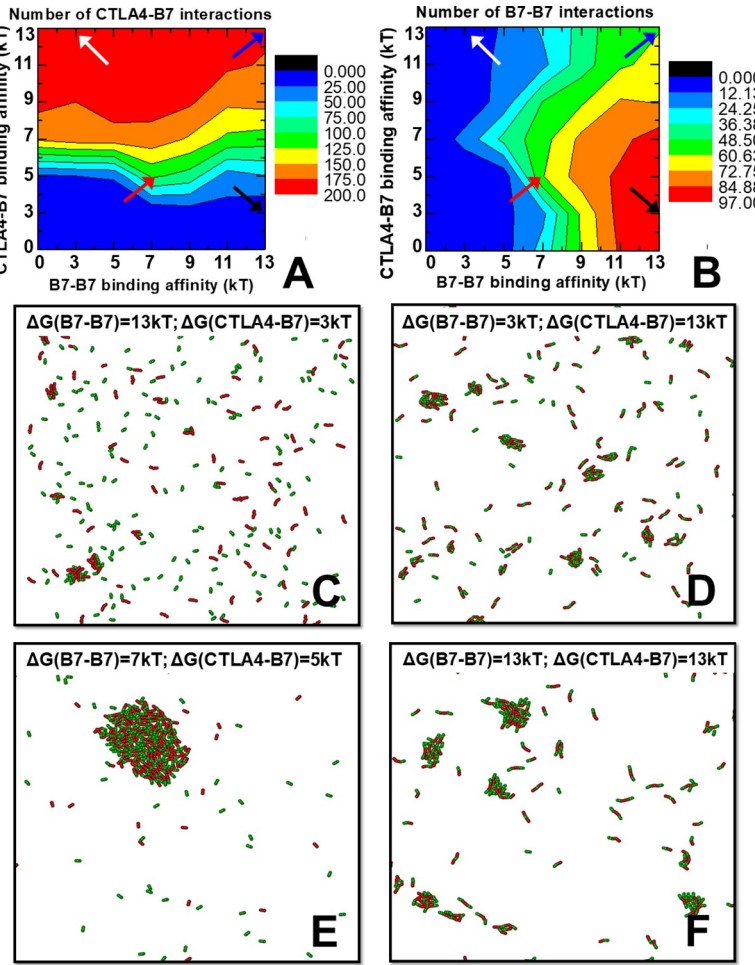

**Fig 3. We systematically changed the binding affinities in both CTLA-4/B7 *trans*-interactions and B7 homo-dimerization from 0 to -13kT.** The contours in the two-dimensional heat maps indicate the number of CTLA-4/B7 *trans*-interactions (**A**) and the number of B7-B7 dimers (**B**), respectively. Detailed color indices are listed on the right-hand sides of each map. The x and y axes represent the values of two binding affinities. The final configurations from some representative areas in the contour maps were selected. The configuration corresponding to the black arrows in the maps is shown in (**C**). The configuration corresponding to the while arrows in the maps is shown in (**D**). The configuration corresponding to the red arrows in the maps is shown in (**E**). Finally, the configuration corresponding to the blue arrows in the maps is shown in (**F**).

dimerization but weak *trans*-interaction, corresponding to the black arrows in **Fig 3A and 3B**. No large oligomers were found in the system due to the lack of strong *trans*-interactions. **Fig 3D** shows the configuration under weak homo-dimerization but strong *trans*-interaction, corresponding to the white arrows in **Fig 3A and 3B**. Similarly, only small oligomers were obtained. These results indicate that the formation of large oligomers requires the participation of both *trans*-interaction and homo-dimerization. Therefore, a configuration with both affinities of *trans*-interaction and homo-dimerization in the medium range was plotted in **Fig 3E**, corresponding to the area in which *trans*-interaction and homo-dimerization are positively coupled (the red arrows in **Fig 3A and 3B**). A single large cluster was formed under this energy combination, indicating this positive coupling is caused by the spatial co-localization of these two types of interactions so that they cannot easily escape from the cluster. Moreover, the large conformational change from small oligomers to large cluster driven by a small increase of

binding affinities suggests that there is a phase transition during the clustering process. Finally, a configuration with strong affinities of both *trans*-interaction and homo-dimerization was plotted in **Fig 3F**, corresponding to the blue arrows in **Fig 3A and 3B**. Comparing with **Fig 3E**, the figure shows that a higher number of smaller clusters were formed under this condition. This result indicates that ligands and receptors are kinetically trapped in small clusters under strong interactions, therefore reducing the positive coupling between *trans*-interaction and homo-dimerization. It is worth mentioning that the same tests were also performed on the systems with lower and higher surface densities of ligands and receptors by changing the size of the simulation box, as summarized in **S1 Fig**. The results under different densities are consistent.

In above study, the ratio of CTLA-4 to B7 is fixed to 1:1. This, however, is likely not the case in real cellular environments. Therefore, additional tests were carried out under different ratios of CTLA-4 to B7. Specifically, three systems were compared. The first system contains 300 B7 monomers and CTLA-4 dimers; the second system contains 200 B7 monomers and 200 CTLA-4 dimers; while the third system contains 100 B7 monomers and 300 CTLA-4 dimers. In each system, the binding affinities of both CTLA-4/B7 *trans*-interaction and B7 homo-dimerization were changed from -3kT to -11kT with an interval of 2kT. A total number of 5×5 = 25 combinations were tested. The overall results are summarized as two-dimensional heat maps in **S3 Fig**. The number of CTLA-4/B7 *trans*-interactions and the number of B7-B7 dimers formed in the first system are shown by **S3A and S3B Fig**, respectively. The number of CTLA-4/B7 *trans*-interactions and the number of B7-B7 dimers formed in the second system are shown by **S3C and S3D Fig**, respectively. The number of CTLA-4/B7 *trans*-interactions and the number of B7-B7 dimers formed in the third system are shown by **S3E and S3F Fig**, respectively. Detailed color indices are listed on the right-hand sides of each map. **S3 Fig** indicates that the patterns in all these systems are very similar. The subtle difference is manifested in the fact that is specified as follows. The number of CTLA-4/B7 interactions under strong binding affinities (upper-right corners of the maps in **S3A, S3C and S3E Fig**) shows the most negative correlation in the first system and the weakest negative correlation in the third system. In contrast, the number of B7 homo-dimers under strong binding affinities (upper-right corners of the maps in **S3B, S3D and S3F Fig**) shows the most negative correlation in the third system and the weakest negative correlation in the first system. This can be explained as follows. If the number of B7 monomers is higher than the number of CTLA-4 dimers, the dimerization of B7 monomers will dominate the system and thus cause stronger impacts on CTLA-4/B7 interactions as reflected by the map in **S3A Fig**. On the other hand, if the number of B7 monomers is lower than the number of CTLA-4 dimers, the CTLA-4/B7 interactions will dominate the system and thus cause stronger impacts on dimerization of B7 monomers as reflected by the map in **S3F Fig**.

In summary, our simulations illustrated that the CTLA-4/B7 ligand-receptor complexes can first form linear oligomers, while these oligomers further align together into two-dimensional clusters. Through this spatial organization, the CTLA-4/B7 *trans*-interaction and the homo-dimerization between B7 ligands will be positively coupled with each other, depending on the strength of their binding affinities.

## The B7/PD-L1 cis-interaction negatively regulates the CTLA-4/B7 oligomerization

We have observed the formation of 2D clusters by the combination of CTLA-4/B7 *trans*-interactions and B7 dimerization. In this section, the other type of ligand PD-L1 was further introduced into the simulation system. As shown in **Fig 1B**, PD-L1 is also presented on the surface

of APC and can form a *cis*-interaction with B7. Moreover, this *cis*-interaction and B7 homo-dimerization compete against each other with the same binding site in B7. In order to investigate the impacts of this B7/PD-L1 *cis*-interaction on the oligomerization of CTLA-4/B7 complexes, we systematically changed the surface density of PD-L1 in the system, as well as the binding affinity of the *cis*-interaction. To avoid further complexity, the surface densities of CTLA-4 and B7 were fixed. The binding affinities of CTLA-4/B7 *trans*-interaction and B7 homo-dimerization also remained unchanged. More specifically, the simulation box has the dimension of 500nm×500nm×20nm. We changed the binding affinities of the B7/PD-L1 *cis*-interactions from -3kT to -13kT, with an interval of 2kT. We also increase the number of PD-L1 in the system from 20 and 220 molecules, with an interval of 40 molecules. As a result, 6×6 = 36 combinations in total were tested. Within the simulation of each combination, 200 CTLA-4 receptor dimers were randomly placed on the surface of T cell as the initial configuration. On the opposite side of APC surface, another 200 B7 were distributed in addition to PD-L1. The affinity of their *trans*-interaction equals 5kT and the affinity of dimerization between two B7 equals 7kT. This combination corresponds to the situation under which CTLA-4 and B7 can form large clusters.

The overall results from the simulations of above combinations are summarized as two-dimensional heat maps in **Fig 4**. Different colors in the maps indicate the number of interactions which types and their corresponding number indices are listed at the right-hand side of the figure. The values of *cis*-binding affinity and the number of PD-L1 in the system are marked along x and y axes, respectively. **Fig 4A** represents the numbers of B7/PD-L1 *cis*-interactions obtained under different combinations. The figure shows that more *cis*-interactions were formed under stronger *cis*-binding affinity and higher surface density of PD-L1, corresponding to the red area in the upper-right corner of the map. **Fig 4B** represents the numbers of B7 dimers obtained under different combinations. Comparing with **Fig 4A**, the figure shows that stronger affinity of B7/PD-L1 *cis*-interactions and higher surface density of PD-L1 resulted in less B7 dimers, corresponding to the blue area in the upper-right corner of the map. This result indicates that the formation of *cis*-interactions with PD-L1 prevents B7 from dimerization, confirming the competition between these two types of interactions. More interestingly, **Fig 4C** represents the numbers of CTLA-4/B7 *trans*-interactions obtained under different combinations. This figure shows that the color contours change from red to blue when they shift from the left-hand side to the right-hand side of the map, suggesting that strong *cis*-interactions between B7 and PD-L1 negatively affect the CTLA-4/B7 *trans*-interactions. We speculate that this negative impact is caused indirectly by the loss of B7 dimerization. This secondary effect also leads to the result that the color contours in **Fig 4C** are more frustrated. Finally, it is worth mentioning that the system containing CTLA-4, B7 and PD-L1 has also been tested by another scenario in which stronger binding affinities (-13kT) were adopted for CTLA-4/B7 *trans*-interaction and B7 homo-dimerization. The same combinations were used to change the PD-L1 density and B7/PD-L1 *cis*-binding affinity. The testing results are summarized in **S2 Fig**. They are consistent with the results shown in **Fig 4**. It is worth mentioning that this negative coupling between CTLA-4/B7 engagement and B7/PD-L1 *cis*-interaction has also been observed by recent experiments [16]. In the experiment, the split nano-luciferase constructs were generated by cloning full-length B7 and PD-L1 into LgBiT or SmBiT vectors. The HEK 293 freestyle cells were co-transfected with these constructs. As a result, an approximately two-fold reduction of the B7 SmBit: PD-L1 LgBit luminescence was obtained in the presence of either soluble CTLA-4 hIgG1 protein or CTLA-4 GFP-expressing cells, confirming our simulation results that CTLA-4/B7 engagement and B7/PD-L1 *cis*-interaction is negatively coupled.

In order to illustrate the detailed kinetic process of the competition between B7/PD-L1 *cis*-interactions and CTLA-4/B7 clustering, two specific systems were selected from all the

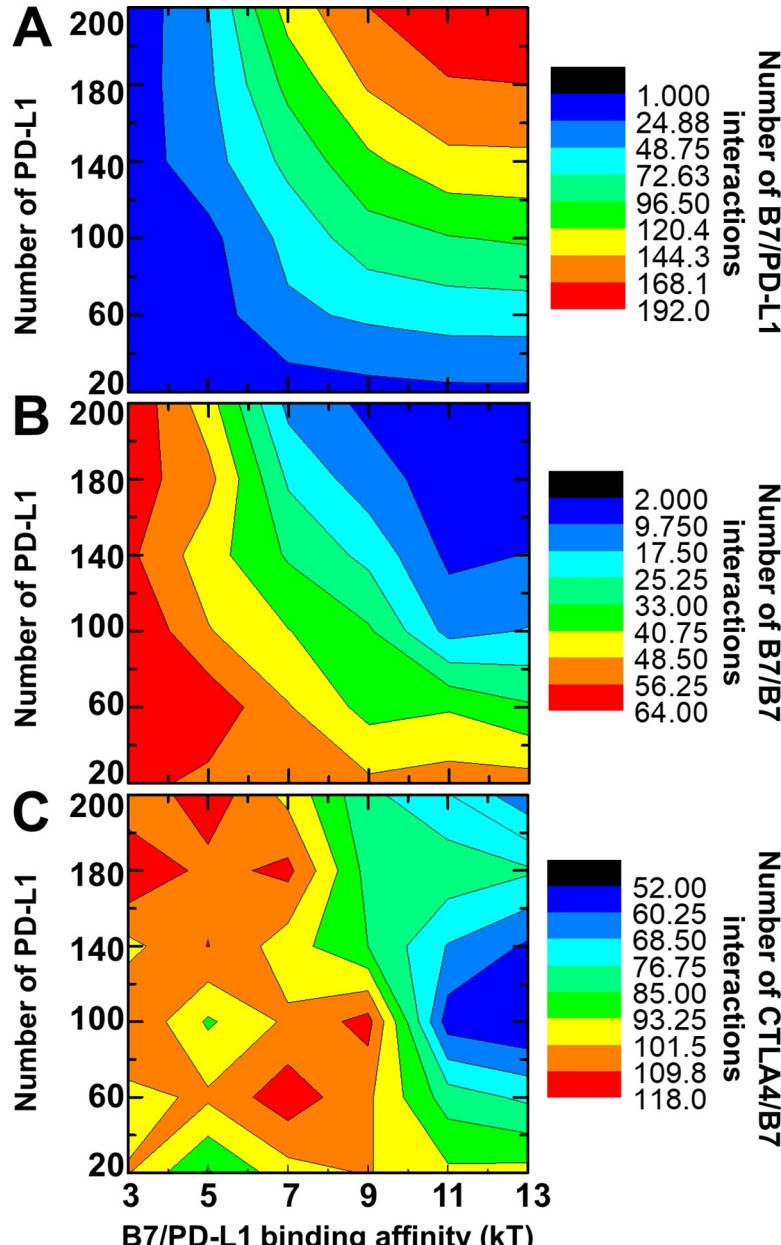

**Fig 4. In addition to CTLA-4 and B7, we further introduced PD-L1 into the simulation system.** We systematically changed the surface density of PD-L1 and the binding affinity of its *cis*-interaction with B7. The tested results are summarized as two-dimensional heat maps. The contours in the maps indicate the number of PD-L1/B7 *cis*-interactions (**A**), the number of B7-B7 dimers (**B**) and the number of CTLA-4/B7 trans-interactions (**C**), respectively. Detailed color indices are listed on the right-hand sides of each map. The x axis represents the values of *cis*-binding affinity, and the y axis indicates the number of PD-L1 on the APC surface.

combinations. The comparison between these two systems is shown in **Fig 5**. In the first system, there were 180 PD-L1 on the surface of APC and a weak binding affinity of -3kT was imposed to the *cis*-interaction between B7 and PD-L1. In the second system, the number of PD-L1 is less (100), but a strong binding affinity of -13kT was imposed to the B7/PD-L1 *cis*-interaction. The changes in the number of B7/PD-L1 *cis*-interactions were plotted in **Fig 5A** as a function of simulation time. The system with the strong binding affinity (red curve) contains

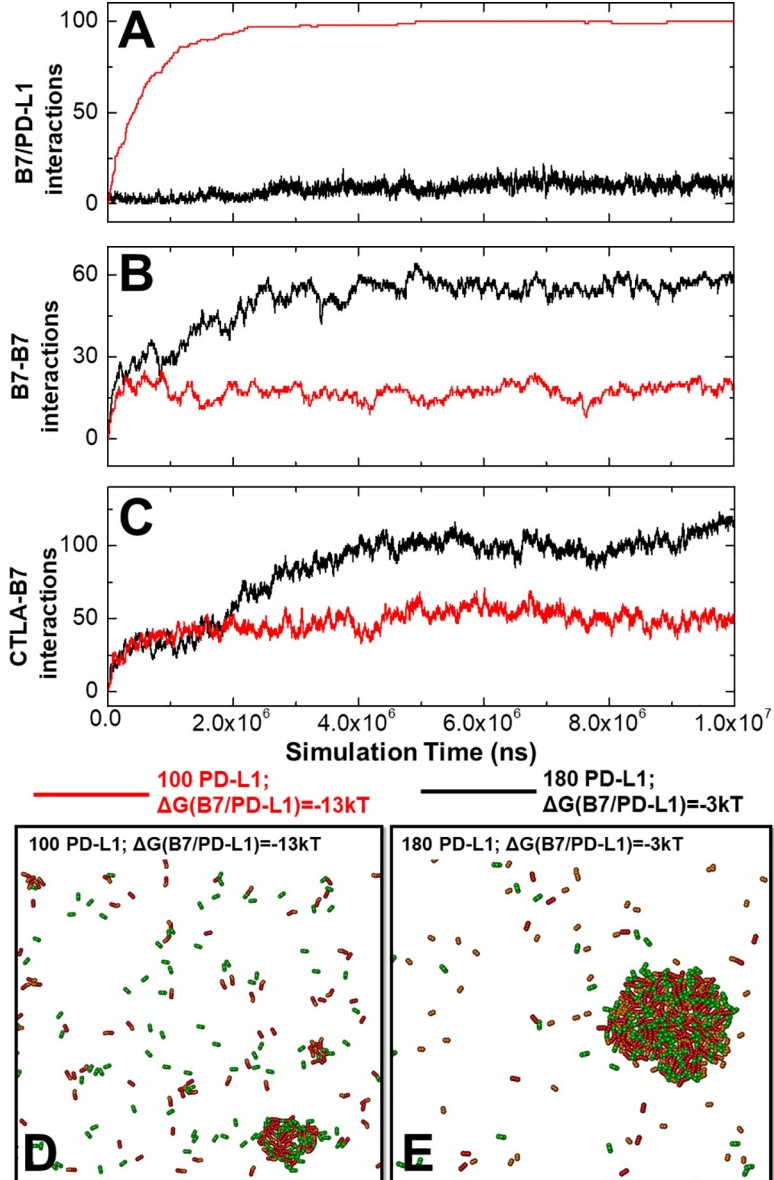

**Fig 5. The kinetic profiles are compared between two selected systems.** The changes of B7/PD-L1 *cis*-interactions as a function of simulation time were plotted in (**A**). The changes of B7 dimers as a function of simulation time were plotted in (**B**). The changes of CTLA-4/B7 *trans*-interactions as a function of simulation time were plotted in (**C**). The system with strong *cis*-binding affinity of -13kT is represented by red curves, while the system with weak *cis*-binding affinity of -3kT is represented by black curves. Finally, the final configurations from these two comparing systems were plotted. The configuration with strong affinity is shown in (**D**) and the configuration with weak affinity is shown in (**E**), respectively.

around 100 *cis*-interactions, much more than the system with the weak affinity (black curve). Almost all PD-L1 proteins were involved in the *cis*-interactions with B7 when its binding affinity equals -13kT. In contrast, only a very limited number of *cis*-interactions (lower than 20) were observed when the affinity equals -3kT, even though the number of PD-L1 was increased from 100 to 180. We further compared the number of B7 homodimers between these two systems in **Fig 5B**. Under weak *cis*-affinity, the black curve in the figure shows that more than 60

homodimers were formed between B7 ligands. However, when the *cis*-affinity increased from -3kT to -13kT, the total number of B7 dimers reduced to less than a half.

Moreover, the numbers of CTLA-4/B7 *trans*-interactions in two systems are compared in **Fig 5C**. Interestingly, different from the red curve which corresponds to the strong *cis*-affinity, the black curve in the figure suggests that the number of CTLA-4/B7 *trans*-interactions under the weak affinity of B7/PD-L1 *cis*-interactions exhibits a two-phase kinetics. The number of *trans*-interactions in the black curve grew at the same rate as the red curve in the first phase. There was a sudden increase in the black curve, however, when the simulation reached $2 \times 10^6$ns, indicating the beginning of the second phase. Finally, the number of *trans*-interactions in the system with weak *cis*-affinity is more than double of the number in the system with strong *cis*-affinity. We speculate that during the first phase, the competition was undergone between B7 homo-dimerization and B7/PD-L1 *cis*-interactions. In the system with weak *cis*-affinity, B7 homo-dimerization dominated the competition. The B7 dimers further induced the formation of more CTLA-4/B7 *trans*-interactions in the second phase. In the system with strong *cis*-affinity, on the other hand, B7 homo-dimerization lost the competition. As a result, the high number of *cis*-interactions indirectly inhibited the further formation of more CTLA-4/B7 *trans*-interactions.

The final configurations from above two systems were plotted in **Fig 5D and 5E**. With the weak binding affinity of B7/PD-L1 *cis*-interaction, a large cluster was formed (**Fig 5E**). The PD-L1 proteins, which are represented in orange, are colocalized together with B7 and CTLA-4 in the same cluster. Within this specific spatial organization, one of the two binding sites in B7 is involved in its *trans*-interaction with CTLA-4, while the other one could be involved in its *cis*-interactions with PD-L1. The low binding affinity of this *cis*-interaction causes the fact that the PD-L1 will easily dissociate from B7 and be replaced by B7 homo-dimerization through the same binding site. The dissociated PD-L1, however, can be recruited to another vacant B7 in the proximal region, leading into the dynamic growth of the cluster. On the other hand, a high number of *cis*-dimers between B7 and PD-L1 are individually distributed in **Fig 5D**. These *cis*-dimers were kinetically trapped by their strong binding affinity and spatially separated from each other. These trapped and separated B7 ligands did not have the opportunity to dimerize and further form linear oligomer with CTLA-4. As a result, a much smaller cluster was obtained under this condition.

In summary, our simulations illustrated that the strong *cis*-interaction between B7 and PD-L1 causes negative impact on the *trans*-interaction between B7 and CTLA-4 through a two-stage kinetic process. The *cis*-interactions prevent B7 proteins from homo-dimerization. As a result, these B7 proteins trap together with PD-L1 and cannot form dynamic aggregation with CTLA-4.

## The balance between CTLA-4 and PD-1 pathways is modulated by their cis-interaction

Except forming *cis*-interaction with B7, the function of PD-L1 is carried out mainly by its engagement with the inhibitory checkpoint receptor PD-1 on the surface of T cells. Therefore, the *trans*-interaction between PD-1 and PD-L1 is finally introduced in this section. As shown in **Fig 1B**, PD-1 and B7 compete against each other with the same binding site in PD-L1. In order to estimate the impacts of this *trans*-interaction alone on the dynamics of the entire network system, we systematically changed the binding affinity between PD-1 and PD-L1. Considering the high number of variables, the values of all the other parameters were fixed in the simulations. Specifically, 200 CTLA-4 dimers and 200 PD-1 receptors were placed on the surface of T cell as random distribution. On the opposite side, 200 B7 and 200 PD-L1 ligands

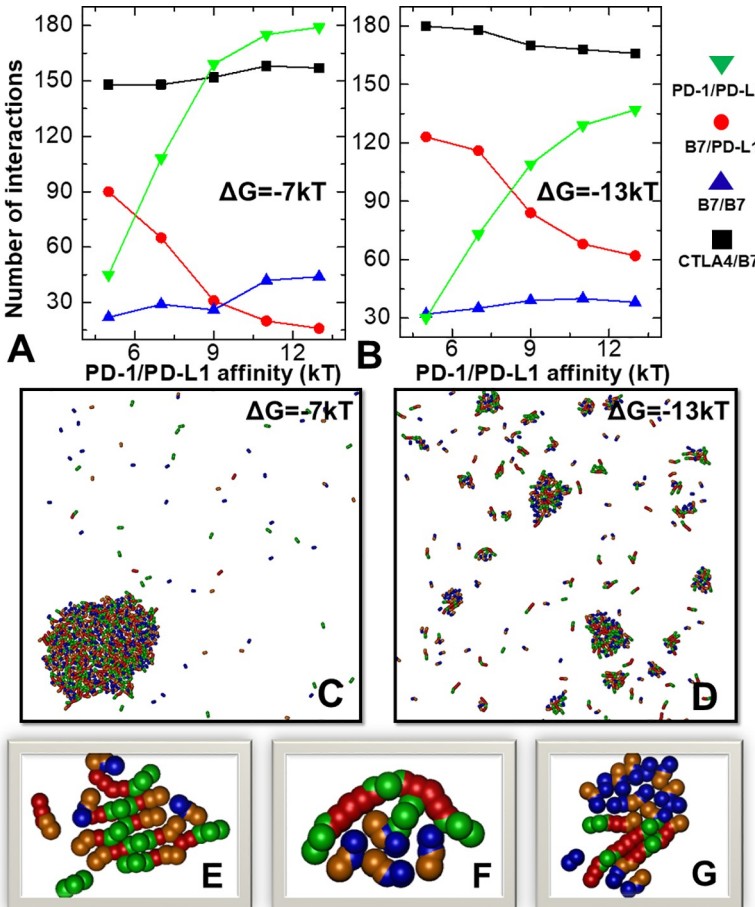

**Fig 6. The dynamics of the system with all four proteins in the network was simulated.** We changed the binding affinities of the PD-1/PD-L1 *trans*-interactions in two scenarios. In the first scenario, moderate binding affinities of -7kT were assigned to all the other interaction, while strong affinities of -13kT were assigned to all the other interactions in the second scenario. The numbers of all types of interactions obtained under different values of PD-1/PD-L1 *trans*-interactions are plotted in **(A)** and **(B)** for the first and second scenarios, respectively. Final configurations from these two scenarios were also plotted in **(C)** and **(D)**. Detailed structure of three clusters formed along the simulation trajectories are displayed in **(E)**, **(F)** and **(G)**.

were randomly distributed on the surface of APC. The simulation box has the dimension of 500nm×500nm×20nm. We changed the binding affinities of the PD-1/PD-L1 *trans*-interactions from -5kT to -13kT, with an interval of 2kT. For the binding affinities of the rest three interactions, two different simulation scenarios were designed. In the first system, a moderate binding affinity of -7kT was assigned to the CTLA-4/B7 *trans*-interaction, the B7 homo-dimerization and the B7/PD-L1 *cis*-interaction, while in the second scenario, a strong affinity of -13kT was adopted to describe these interactions.

The overall test results were plotted in **Fig 6**. The numbers of PD-1/PD-L1 *trans*-interactions formed in the first scenario are shown by green triangles in **Fig 6A** as a function of their binding affinity, while the numbers of B7/PD-L1 *cis*-interactions are shown by red circles. The figure indicates that more PD-L1 bound to PD-1 under stronger affinity of their interaction, leading into less *cis*-interaction formed between B7 and PD-L1. Moreover, the numbers of B7 homodimers increased with the loss of *cis*-interactions, as shown by blue triangles in the figure. In other words, when more PD-L1 ligands are involved in the engagement with their receptors

on T cell, they are less likely to interact with B7 on the same surface, providing the higher probability to B7 homo-dimerization. The increasing number of homodimers between B7, in turn, positively affects their *trans*-interactions with CTLA-4 through the linear oligomerization, as shown by black square in **Fig 6A**. The simulation results from the first scenario suggest that the B7/PD-L1 *cis*-interaction generates a positive correlation between PD-1 and CTLA-4 pathways. The conclusion from the second simulation scenario, on the other hand, is slightly different, which is summarized in **Fig 6B**. Same as the first scenario, the figure shows more PD-1/PD-L1 *trans*-interactions, less B7/PD-L1 *cis*-interactions and more B7 homodimers when the affinity between PD-1 and PD-L1 became stronger. However, under strong interactions between proteins in the network, the second scenario suggests a negative correlation between the numbers of PD-1/PD-L1 and CTLA-4/B7 interactions. Altogether, our simulations indicate that the dynamic property of crosstalk between PD-1 and CTLA-4 pathways depends on the energetics of their molecular interactions.

A representative final configuration was selected for each of the two scenarios. Under moderate binding affinities, the system forms a single large cluster, as shown in **Fig 6C**. Under strong binding affinity, on the other hand, a larger number of smaller clusters were obtained (**Fig 6D**). All four types of proteins were co-localized in these clusters. The close-up view of several clusters formed along the simulation trajectories were further selected and plotted in **Fig 6E–6G**. The configurations of clusters in these figures provide insights about how these ligands and receptors aggregated together. We speculate that the clustering process could be described as following. Firstly, the clusters consist of linear oligomers of CTLA-4 and B7, as we discussed before. The expansion of these linear oligomers can be terminated by PD-L1 through the *cis*-interactions between PD-L1 and B7. As a result, a high proportion of PD-L1 molecules appeared at the ends of CTLA-4/B7 oligomers. Although they can dissociate from the oligomers, their diffusions at cellular interface were restricted due to the geometric confinement of neighboring oligomers. After they captured their receptors PD-1, these *trans*-dimers also formed at the spatial proximal regions. Moreover, this enhancement of local concentration can trap the diffusions of more proteins, which cause the further growth of the entire clusters.

Additionally, we have performed another computational experiment to investigate the role of B7/PD-L1 *cis*-interaction in regulating the balance between CTLA-4 and PD-1 signaling pathways. Two different simulation scenarios were specifically designed. In the first system, 200 CTLA-4 dimers and 200 PD-1 receptors were placed on the surface of T cell as random distribution. On the opposite side, 200 B7 and 200 PD-L1 ligands were randomly distributed on the surface of APC. The simulation box has the dimension of 500nm×500nm×20nm. The binding affinity of CTLA-4/B7 *trans*-interaction equals -5kT, while the binding affinities of B7 homo-dimerization and PD-1/PD-L1 trans-interaction equal -7kT. The binding affinity of B7/PD-L1 *cis*-interaction equals -9kT. In the second scenario, a mutation was introduced to turn off the *cis*-interaction to 0kT. The values of all the other parameters remained unchanged. While the second scenario is referred as mutant (MT) system, the first scenario is referred as wild-type (WT) system. For both MT and WT systems, simulations with the length of $1\times10^7$ns were carried out. We counted the numbers of interactions formed within the last $2\times10^6$ns of both simulations. These interactions form distributions as shown by the box-whisker plots in **Fig 7**.

The distributions of different interactions derived from the MT system are represented by red boxes, while the distributions from the WT system are represented by green boxes. In specific, the distributions of *cis*-interaction between B7 and PD-L1 are compared in **Fig 7A**. No *cis*-interaction was formed in MT system, while an average number of 120 B7/PD-L1 *cis*-dimers were obtained in WT system. Subsequently, the distributions of PD-1/PD-L1 *trans*-interactions, B7 dimerization, and CTLA-4/B7 *trans*-interactions are respectively compared in

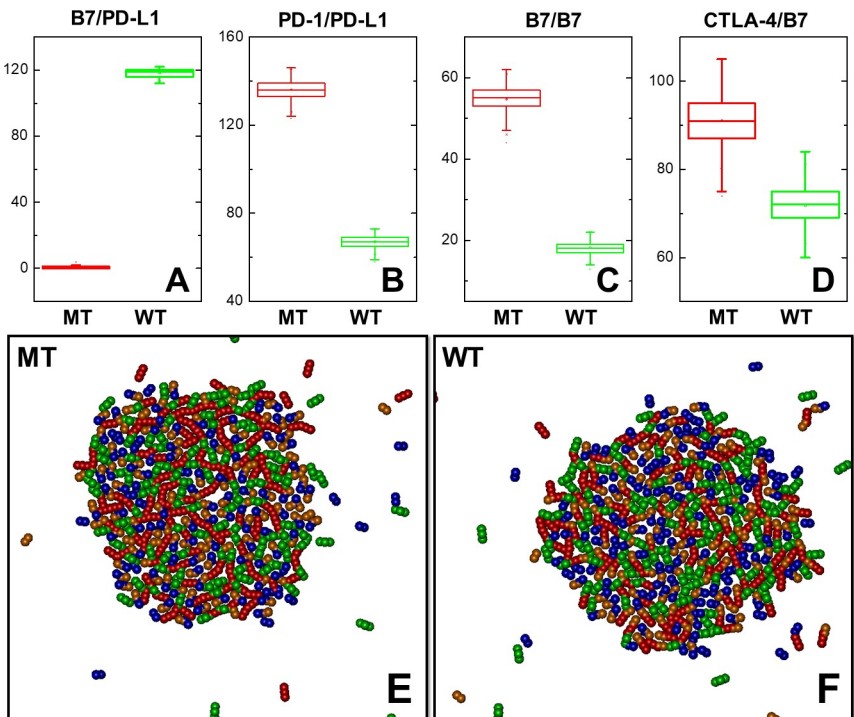

**Fig 7. We have performed a computational experiment in the *cis*-interactions between B7 and PD-L1 was turn on and off in two separate simulation scenarios.** We collected the numbers of different interactions along the last $2\times10^6$ns of both simulation trajectories. Their distributions are compared with each other as box-whisker plots in (**A**), (**B**), (**C**), and (**D**). The boxes in these plots give the range from 25 to 75 percentiles for the number of interactions, while their average number of interactions is marked in the middle of each box. The whisker indicates the outlier of the distribution with the coefficient equal 1.5. The type of interactions is indicated on top of each plot. The system with *cis*-interaction (WT) is shown by green boxes, while the system without *cis*-interaction (MT) is shown by red boxes. Finally, the close-up view of the largest cluster formed in the final configuration of the MT scenario is shown in (**E**), and the close-up view of the largest cluster formed in the final configuration of the WT scenario is shown in (**F**).

**Fig 7B–7D**. The figures indicate that for these three types of interactions, their average numbers formed in the WT systems (green boxes) are all lower than the MT system (red boxes). Considering that the only difference between the WT and MT systems is the binding affinity of *cis*-interaction between B7 and PD-L1, our statistical results thus suggest that the signaling of both co-inhibitory receptors CTLA-4 and PD-1 can be negatively regulated by the lateral crosstalk between their ligands (B7 and PD-L1). This conclusion is supported by previous experimental evidences [15,54]. In the experiments, Raji B cells were first stained by CTLA-4-Fc in a dose dependent manner. It was found that co-expression of PD-L1 with 5.9-fold excess to B7 significantly reduced CTLA-4-Fc staining, leading into an approximately 4.7-fold higher dissociation constant. This result suggests that the original B7/CTLA-4 interaction can be inhibited by the *cis*-interaction between B7 and PD-L1. Vice versa, when PD-L1 transduced, CD80 knockout Raji B cells were stained by PD-1-Fc in a concentration dependent manner, this PD-1-Fc staining was dramatically decreased by co-expression of CD80 with 3.5-fold excess of PD-L1. This result suggests that the PD-1/PD-L1 *trans*-interaction can also be inhibited by the *cis*-interaction between B7 and PD-L1. The close-up views of the largest cluster formed in the final configuration of two scenarios are shown in **Fig 7E and 7F**. All four types of proteins are co-localized together in both systems. However, there are still subtle differences between MT and WT systems. Firstly, more PD-1 receptors (blue) are engaged with PD-L1 (orange) in MT system (**Fig 7E**) than WT system (**Fig 7F**). Secondly, more homodimers

between B7 (red) can be found in MT system than WT system. As a result, more dimerized B7 ligands are further involved in the oligomerization with CTLA-4 (green) in MT system, while these oligomers are more frequently terminated by the PD-L1 through its *cis*-interactions with B7. These observations are consistent with the statistical results described earlier.

In summary, using the simple network motif as a testing system for co-inhibitory immune complex assembly, we demonstrated that both co-inhibitory signaling pathways activated by B7 and PD-L1 can be down-regulated by the potential *cis*-interaction between these two ligands. Moreover, the dynamic and the spatial properties of the immune complex assembly are highly determined by the energetics of molecular interactions in the network.

## Concluding discussion

Like many other systems of multi-cellularity, it has become increasingly appreciated that ligand-receptor interactions at the interface between T cells and APCs are more promiscuous than we used to think [55]. The functional role of this binding promiscuity in regulating the elaborate dynamics of adaptive immune response, however, is not understood. One example is the gradually revealed complexity underlying the binding of co-regulatory receptors on T cells. Binding of receptors PD-1 and CTLA-4 with their corresponding ligands PD-L1 and B7 activates two central co-inhibitory signaling pathways that immune system uses to suppress T cell functions. Interestingly, recent experiments have identified a new interaction between PD-L1 and B7. Further evidences demonstrated that this interaction occurs in *cis* on the surface of APC, suggesting that the two co-inhibitory receptors and their ligands can be assembled into dynamic complexes. Inspired by these studies, as well as the structural information of relevant ligand-receptor complexes, a coarse-grained model was developed to characterize the dynamics and spatial organization of this immune complex assembly. Diffusions of ligands and receptors are confined at cellular interfaces to present a realistic intercellular environment. The spatial arrangement of multiple binding sites in each molecule has also been captured by this model. Our simulations show that the PD-L1/B7 *cis*-interaction affects the *trans*-interaction between CTLA-4 and B7 indirectly by reducing the homo-dimerization between B7. The competition for PD-L1 binding between B7 and PD-1 can relief this effect. This new *cis*-interaction between PD-L1 and B7, therefore, regulates the balance between the two classic *trans*-interactions of PD-1/PD-L1 and CTLA-4/B7. We also illustrated that the ligand-receptor complexes can aggregate into large clusters, which sizes are dependent on the strength their interactions. We suggest that these clusters can serve as a spatial platform to maintain the dynamics of a network consisting of intersecting and competing interactions among co-regulatory receptors and their ligands. In summary, this computational study offers new insights to our mechanistic understanding of co-regulation in immune system.

Further improvements and extension of current model will be made, so that it can be used to study more complicated scenarios. First of all, the linkers between different representative groups of a protein are fixed in current coarse-grained model, while the cell membranes are modeled as plain surfaces. In other words, the deformation of lipid bilayer and the fluctuations along intramolecular degrees of freedom are neglected for simplification. These dynamic factors can be explicitly taken into account as our future model extension. For instance, the intramolecular conformational fluctuations can be modeled by using a harmonic potential to describe the deformation between different representative groups. Similarly, plasma membrane will be modeled as an elastic medium. Secondly, the binding constants mediating the *trans* and *cis*-interactions between ligands and receptors are parameters that were predetermined in the simulations. These parameters can be estimated more quantitatively through computational methods with higher resolutions, if their experimental measurements are not

available. For an example, binding rates and affinities of two interacting proteins can be calculated by simulations with their structural details and using physics-based or knowledge-based scoring functions to describe their interactions [56]. The calculated binding constants can further be integrated into current model by a multiscale modeling framework [57], so that the structural and energetic features at the binding interfaces of an immune complex can be explicitly factored in. Finally, our method not only can be applied to study natural proteins, but also can introduce soluble biotherapeutics into simulations. For instance, nivolumab and ipilimumab are two FDA approved biologics for cancer immunotherapy [58]. They work as checkpoint inhibitors through their binding to the extracellular domains of PD-1 and CTLA-4, respectively. Clinical evidences showed the favorable outcomes of their combination therapy in the treatment of advanced melanoma [59]. As a result, we hypothesize that a new design of biologics which covalently links these two antibodies together will be able to binding PD-1 and CTLA-4 on the surface of the same T cells with higher specificity, due to its higher binding avidity. This hypothesis can be testified by extending the simulation framework of current study. The outputs will help the development of new drugs with higher efficacy and lower risk for off-target side effects.

## Supporting information

**S1 Fig. We systematically changed the binding affinities in both CTLA-4/B7 *trans*-interactions and B7 homo-dimerization from 0 to -13kT under different concentration.** The contours in the two-dimensional heat maps indicate the number of CTLA-4/B7 *trans*-interactions **(A)** and the number of B7-B7 dimers **(B)** formed in the system with the lower concentration. The contours in the two-dimensional heat maps indicate the number of CTLA-4/B7 *trans*-interactions **(C)** and the number of B7-B7 dimers **(D)** formed in the system with the higher concentration. Detailed color indices are listed on the right-hand sides of each map. The x and y axes represent the values of two binding affinities.
(TIF)

**S2 Fig. Additional tests were carried out to the system consisting of CTLA-4, B7 and PD-L1, in which stronger binding affinities (-13kT) were adopted for CTLA-4/B7 *trans*-interaction and B7 homo-dimerization.** We systematically changed the surface density of PD-L1 and the binding affinity of its *cis*-interaction with B7. The tested results are summarized as two-dimensional heat maps. The contours in the maps indicate the number of PD-L1/B7 *cis*-interactions **(A)**, the number of B7-B7 dimers **(B)** and the number of CTLA-4/B7 trans-interactions **(C)**, respectively. Detailed color indices are listed on the right-hand sides of each map. The x axis represents the values of *cis*-binding affinity, and the y axis indicates the number of PD-L1 on the APC surface.
(TIF)

**S3 Fig. Different ratios of CTLA-4 to B7 were tested.** Under each ratio, the binding affinities of both CTLA-4/B7 trans-interaction and B7 homo-dimerization were changed from -3kT to -11kT with an interval of 2kT. The number of CTLA-4/B7 trans-interactions and the number of B7-B7 dimers formed in the system containing 300 B7 monomers and 100 CTLA-4 dimers are shown in **(A)** and **(B)** as 2D colorful heat map. Different combinations of binding affinities are indexed by the y and x aces of the maps. Similarly, the number of CTLA-4/B7 trans-interactions and the number of B7-B7 dimers formed in the system containing 200 B7 monomers and 200 CTLA-4 dimers are shown by **(C)** and **(D)**. Finally, the number of CTLA-4/B7 trans-interactions and the number of B7-B7 dimers formed in the system containing 100 B7 monomers and 300 CTLA-4 dimers are shown by **(E)** and **(F)**. Detailed color indices are listed on

the right-hand sides of each map.
(TIF)

## Acknowledgments

We acknowledge that all computational supports were provided by Albert Einstein College of Medicine High Performance Computing Center.

## Author Contributions

**Conceptualization:** Yinghao Wu.

**Data curation:** Zhaoqian Su, Yinghao Wu.

**Formal analysis:** Zhaoqian Su, Kalyani Dhusia, Yinghao Wu.

**Funding acquisition:** Yinghao Wu.

**Investigation:** Zhaoqian Su, Kalyani Dhusia, Yinghao Wu.

**Methodology:** Zhaoqian Su, Yinghao Wu.

**Project administration:** Yinghao Wu.

**Resources:** Zhaoqian Su, Kalyani Dhusia.

**Software:** Zhaoqian Su, Yinghao Wu.

**Supervision:** Yinghao Wu.

**Validation:** Zhaoqian Su, Kalyani Dhusia.

**Visualization:** Zhaoqian Su, Yinghao Wu.

**Writing – original draft:** Yinghao Wu.

**Writing – review & editing:** Yinghao Wu.

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
