## [Decision Letter · Decision Letter 0]

17 Dec 2020

Dear Dr. Wu,

Thank you very much for submitting your manuscript "A Computational Study of Co-inhibitory Immune Complex Assembly at the Interface between T cells and Antigen Presenting Cells" for consideration at PLOS Computational Biology.

As with all papers reviewed by the journal, your manuscript was reviewed by members of the editorial board and by several independent reviewers. In light of the reviews (below this email), we would like to invite the resubmission of a significantly-revised version that takes into account the reviewers' comments.

We cannot make any decision about publication until we have seen the revised manuscript and your response to the reviewers' comments. Your revised manuscript is also likely to be sent to reviewers for further evaluation.

Sincerely,

Guanghong Wei

Associate Editor

PLOS Computational Biology

Nir Ben-Tal

Deputy Editor

PLOS Computational Biology

Reviewer's Responses to Questions

**Comments to the Authors:**

Reviewer #1: In the paper "A Computational Study of Co-inhibitory Immune Complex Assembly at

the Interface between T cells and Antigen Presenting Cells", Su et al develop a coarse-grained mesoscale model of protein-ligand and protein-protein interactions in order to elucidate their mechanisms of assembly in T cells.

The main novelty of the approach relies in accounting for different binding interfaces and kinetic rates, which give rise to different steady state properties of the system. The author carry on a systematic parameter scan in order to evaluate the relative importance on these parameters on the amount and relative types of interactions.

While the paper is interesting and the result are novel I recommend three minor modifications:

1) a more systematic and quantitative comparison of the obtained results with experiments. This comparison is often implied, but a quantitative description would strengthen the conclusions.

2) A careful editing of the paper. A number of sentences would benefit from a grammar check.

3) A few general sentences in the introduction about the mechanisms of T cells assembly at interface would better familiarize the reader with the specific topic of the paper.

Reviewer #2: The manuscript by Su et al. reports a coarse-grained reaction-diffusion simulation model of T cell receptors, CTLA-4 and PD-1, and their corresponding antigen presenting cell (APC) ligands, B7 and PD-L1. By varying the binding constants in the model, the authors showed that cis-interactions between the two ligands on the APC membrane can inhibit CTlA-4/B7 oligomerization because of the competition for an overlapped B7 binding and homodimerization site. The introduction of PD-1 into the system reduces the inhibition by competing for the PD-L1 binding site that is shared with B7. The authors suggest these competitive bindings can regulate the signalling pathways activated by B7 and PD-L1.

To be acceptable for publication, the authors should address the following issues:

1. The simulation model assumes that a dimer of CTLA-4 can bind to two B7 monomers simultaneously. Reference to previously reported structural or other experimental evidence that support this assumption should be provided.

2. The translational diffusion coefficient of the ligand-receptor complex was set to 5 um2/s but for ligand-ligand hetero- and homo-dimers, the coefficient is 0 um2/s. Why the discrepancy? Would letting all dimers, whether they are ligand-receptor or ligand-ligand, diffuse at 5 um2/s change any of the results presented in the manuscript?

3. In the model, the ratio of CTLA-4 to B7 is always fixed to 1:1, which is likely not the case in vivo. The authors should show how does the correlation between CTLA-4/B7 trans-interaction and B7 homodimerization in Figure 3 change when the ratio is changed.

There were some language issues, for example:

page 6: "Incorporate these factors into.."

page 15: "-3kT, even the number..."

page 17: "In another word,..."

page 28: Caption of Figure 4: "In addition to CTLA-4 and B7, ligand PD-L1, we further introduced PD-L1 into the simulation system."

**Have all data underlying the figures and results presented in the manuscript been provided?**

Reviewer #1: None

Reviewer #2: **No: **Ideally, the software and simulation model should be made accessible to reproduce the results.

PLOS authors have the option to publish the peer review history of their article (what does this mean?). If published, this will include your full peer review and any attached files.

Reviewer #1: No

Reviewer #2: No
---

## [Decision Letter · Decision Letter 1]

21 Feb 2021

Dear Dr. Wu,

We are pleased to inform you that your manuscript 'A Computational Study of Co-inhibitory Immune Complex Assembly at the Interface between T cells and Antigen Presenting Cells' has been provisionally accepted for publication in PLOS Computational Biology.

Best regards,

Guanghong Wei

Associate Editor

PLOS Computational Biology

Nir Ben-Tal

Deputy Editor

PLOS Computational Biology

Reviewer's Responses to Questions

**Comments to the Authors:**

Reviewer #1: The authors have successfully addressed all my concerns and I recommend publication

Reviewer #2: The authors have revised the paper according to reviewers' comments. They have clarified points raised and performed new simulations reported in new graphs and referenced more relevant literature. The response of the authors to all the previous recommendations is satisfactory. I am happy to recommend the paper for publication as it is.

**Have all data underlying the figures and results presented in the manuscript been provided?**

Reviewer #1: None

Reviewer #2: **No: **I could not find numerical data that underlies graphs in a public repository or in spreadsheet form as supporting information.

PLOS authors have the option to publish the peer review history of their article (what does this mean?). If published, this will include your full peer review and any attached files.

Reviewer #1: No

Reviewer #2: No

---

## [Editor Report · Acceptance letter]

2 Mar 2021

PCOMPBIOL-D-20-01916R1 

A Computational Study of Co-inhibitory Immune Complex Assembly at the Interface between T cells and Antigen Presenting Cells

Dear Dr Wu,

I am pleased to inform you that your manuscript has been formally accepted for publication in PLOS Computational Biology. Your manuscript is now with our production department and you will be notified of the publication date in due course.

With kind regards,

Alice Ellingham
